# Antiviral Effects of *Houttuynia cordata* Polysaccharide Extract on Murine Norovirus-1 (MNV-1)—A Human Norovirus Surrogate

**DOI:** 10.3390/molecules24091835

**Published:** 2019-05-13

**Authors:** Dongqing Cheng, Liang Sun, Songyan Zou, Jiang Chen, Haiyan Mao, Yanjun Zhang, Ningbo Liao, Ronghua Zhang

**Affiliations:** 1Department of Nutrition and Food Safety, Zhejiang Provincial Center for Disease Control and Prevention, Hangzhou 310006, China; chengdq@zcmu.edu.cn (D.C.); lsun@cdc.zj.cn (L.S.); jchen@cdc.zj.cn (J.C.); hymao@cdc.zj.cn (H.M.); yjzhang@cdc.zj.cn (Y.Z.); 2College of Medical Technology, Zhejiang Chinese Medical University, Hangzhou 310053, China; zou_songyan@126.com; 3School of Public Health, Division of Infectious Diseases and Vaccinology, University of California, Berkeley, CA 94720, USA

**Keywords:** antiviral effects, *Houttuynia cordata*, polysaccharide, water extract, ethanol extract, murine norovirus-1

## Abstract

*Houttuynia cordata* is an herbal plant rich in polysaccharides and with several pharmacological activities. Human noroviruses (HuNoVs) are the most common cause of foodborne viral gastroenteritis throughout the world. In this study, *H. cordata* polysaccharide (HP), with a molecular weight of ~43 kDa, was purified from *H. cordata* water extract (HWE). The polysaccharide HP was composed predominantly of galacturonic acid, galactose, glucose, and xylose in a molar ratio of 1.56:1.49:1.26:1.11. Methylation and NMR analyses revealed that HP was a pectin-like acidic polysaccharide mainly consisting of α-1,4-linked Gal*p*A, β-1,4-linked Gal*p*, β-1,4-linked Glc*p*, and β-1,4-linked Xyl*p* residues. To evaluate the antiviral activity of *H. cordata* extracts, we compared the anti-norovirus potential of HP with HWE and ethanol extract (HEE) from *H. cordata* by plaque assay (plaque forming units (PFU)/mL) for murine norovirus-1 (MNV-1), a surrogate of HuNoVs. Viruses at high (8.09 log10 PFU/mL) or low (4.38 log10 PFU/mL) counts were mixed with 100, 250, and 500 μg/mL of HP, HWE or HEE and incubated for 30 min at room temperature. *H. cordata* polysaccharide (HP) was more effective than HEE in reducing MNV-1 plaque formation, but less effective than HWE. When MNV-1 was treated with 500 μg/mL HP, the infectivity of MNV-1 decreased to an undetectable level. The selectivity indexes of each sample were 1.95 for HEE, 5.74 for HP, and 16.14 for HWE. The results of decimal reduction time and transmission electron microscopic revealed that HP has anti-viral effects by deforming and inflating virus particles, thereby inhibiting the penetration of viruses in target cells. These findings suggest that HP might have potential as an antiviral agent in the treatment of viral diseases.

## 1. Introduction

*Houttuynia cordata* Thunb. is a medicinal plant commonly found in Southeast Asia. In China, the young roots and green leaves of *H. cordata* are popular vegetable products, being used in the preparation of beverages by boiling decoction. *Houttuynia cordata* contains a wide range of compounds including polysaccharides, fatty acids, polyphenols, flavonoids, and sterols, and has anti-viral, antifungal, detoxifying, and anti-bacterial properties [1]. Hayashi et al. [2] reported that *H. cordata* extract reduces the infectivity (by >4 log) of several viruses, including the influenza virus and HIV. Even though several biological activities of *H. cordata* have been studied [3], the effects of *H. cordata* against human noroviruses (HuNoVs) and its mechanism of action remain unknown.

Human noroviruses, which belong to the Caliciviridae family, are the most common cause of foodborne viral gastroenteritis throughout the world. Human noroviruses can be divided into five major genogroups, genogroup I (GI) through genogroup V (GV); Genogroup II.4 (GII.4) HuNoVs are closely related to most foodborne noroviral outbreaks. Human noroviruses infections mainly occur via person-to-person transmission through fecal–oral route or by consuming contaminated food. Human noroviruses are highly tolerant to environmental changes and have low infectious doses of eight to 10 viral particles [4]. Most sources of foodborne disease caused by HuNoVs are foods prepared with contaminated hands or cookware. Currently, there is no commercially available antiviral drug or vaccine for the prevention of norovirus infections or outbreaks.

The inactivation of HuNoVs mainly relies on physical treatments including chemical methods (e.g., titanium dioxide, sodium hypochlorite, and hydrogen peroxide), heat treatment, and ultraviolet radiation. However, the use of such treatments is frequently inadequate in the food industry. Therefore, it is of utmost importance to investigate safe anti-noroviral agents that can be consumed. Several compounds isolated from plants (e.g., polysaccharides, polyphenols, and flavonoids) have antimicrobial activity [5]. A recent review suggested that plant extracts containing polysaccharides and polyphenols, such as *Ganoderma lucidum* polysaccharide (GLPS), Pericarpium granati extract (PGE), and Pomegranate extract (PE), might prevent infection from NoV surrogates [6]. These natural products are safe antiviral agents because they can be eaten, and their nutrition-promoting properties have long been studied [7,8].

Even though it has been reported that HuNoVs may be cultured in B cells in vitro, there are several disadvantages to this method for screening natural antiviral compounds [4]. Anti-noroviral effects are evaluated from the reduction in infectivity of cultivatable HuNoV surrogates such as murine norovirus (MNV-1), bacteriophage MS2, or other in vitro models of HuNoV infectivity detection (in situ capture qRT-PCR) [9,10,11]. Among them, MNV-1, a genogroup V (GV) cultivable norovirus, is currently recognized as the most suitable surrogate for HuNoVs [10].

In the study, we identified the anti-norovirus potential of crude water extract (HWE), purified polysaccharide (HP), and ethanol extract (HEE) from *H. cordata* by plaque assay for MNV-1. Additionally, we investigated the structure, chemical composition, and antiviral mechanism of HP.

## 2. Results

### 2.1. Chemical Analyses of HP, HWE, and HEE

Extraction yields of HP, HWE, and HEE from *H. cordata* were 5.7%, 16.3%, and 9.6%, respectively. The chemical compositions of the extracts are shown in Table 1. Ethanol extract samples contained 25.42 ± 2.31 gallic acid equivalents (GAE)/mg total phenolic, 19.76 ± 3.73 retinol equivalent (RE)/mg total flavonoid, and 2.37 ± 1.45% neutral sugars. Crude water extract samples contained 14.59 ± 2.41 GAE/mg total phenolic, 10.62 ± 1.43 RE/mg total flavonoid, 25.67 ± 1.97% neutral sugars, and 12.34 ± 2.01% uronic acids. Compared to HEE, HWE contained more carbohydrate. The total carbohydrate content (38.01%) of HWE was calculated by adding neutral sugar and uronic acid values. Purified polysaccharide was isolated from HWE. The HPLC profile of HP revealed a single peak with a molecular weight (Mw) of 43 kDa. The total carbohydrate content of HP was 81.12%. Monosaccharides in HP were mainly galacturonic acid (GlaUA), xylose (Xyl), glucose (Glc), and galactose (Gal). Minor amounts of glucuronic acid (GlcUA), arabinose (Ara), mannose (Man), and rhamnose (Rha) were obtained. The protein content of HP was 0.89%. The sugar composition of HP was similar to the *H. cordata* polysaccharide fractions (HCP and HCP-2) reported in past studies [12,13].

The IR spectrum of HP from *H. cordata* is shown in Figure 1. The band at 3406.0 cm^−1^ is attributed to O–H or N–H stretching vibration. The band at 2931.3 cm^−1^ is attributed to sp^3^ C–H. The peak at 1644.8 cm^−1^ might be due to amide C=O stretching vibration and/or C=C stretching vibration and/or N–H [14]. The bands at 1644.8 and 1403.0 cm^−1^ are attributed to the presence of proteins in HP. Moreover, the characteristic absorption at 1200–1000 cm^−1^ is attributed to the presence of glycosidic linkages C–O and/or C–N in HP [15]. All these results suggest that HP is an acidic polysaccharide.

The methylation analysis results of HP are summarized in Table 2. The polysaccharide HP mainly consisted of 1,4-linked Gal*p*A, 1,4-linked Gal*p*, 1,4-linked Glc*p*, and 1,4-linked Xyl*p* residues in a molar ratio of 1.74:0.93:0.95:0.84. The high prevalence of Gal*p* and Gal*p*A residues revealed that HP might be a pectin-like acidic polysaccharide with a 1,4-linked Gal*p* core. In addition, other small proportions of residues 1,5-linked Araf, 1,2,4-linked Rhap, 1,6-linked Gal*p*, 1,4,6-linked Gal*p*, and terminal-linked Gal*p* and Glc*p*, which have been reported in the side chains, were detected.

Most of the β-anomeric protons are in the δ4–5 ppm range while most of the α-anomeric protons usually appear in the δ5–6 ppm region [16]. The resonances ranging from 4.8 to 5.24 ppm in the ^1^H-NMR spectrum of HP (Figure 2A) revealed that the sugar residues of HP might be connected by β- and α-glycosidic bonds. The anomeric protons from each monosaccharide can give recognizable signals depending on their β- or α-configurations. The signal at δ5.04, 3.74, 3.97, 4.38, and 4.68 ppm was clearly assigned to H-1 to H-5 of α-D-GalA residues, respectively. In a ^13^C-NMR spectrum, the signals derived from α-anomeric carbons will appear in the range δ95–101 ppm while most of the β-anomeric carbons usually appear in the δ101–105 ppm region [16]. The ^13^C-NMR spectrum of HP (Figure 2B) showed the anomeric peaks were centralized between δ98.28 and δ105.06 ppm, indicating two kinds of anomeric configurations for monosaccharide residues of HP. The ^13^C-NMR spectrum showed characteristic anomeric signals at δ99.7, 103.1, and 104.4 ppm, which were due to the C1 resonances of 1,4-α-d-GalA residues, 1,4-β-d-Xylp residues, and 1,4-β-d-Glcp residues, at δ105.5 ppm due to the C1 resonances of β-d-Galp residues (Appendix A). Other predominant signals at δ70.9, 71.4, 80.5, 73.4, and 176.4 ppm were related to C-2 to C-6 of α-d-GalA residues, respectively. In addition, the signals at the low field from δ160 to δ180 ppm in the ^13^C-NMR spectrum of HP (Figure 2B) illustrated that the polysaccharide contained uronic acid [17]. All the NMR chemical shifts were compared with previously reported values [18,19].

### 2.2. Cytotoxicity Assay

The cytotoxicity assay was determined by calculating CC_50_. The effects of *H. cordata* extracts on the survival rate of the murine macrophage cell lines (RAW 264.7) are shown in Figure 3. Ethanol extract, HWE, and HP had no severe cytotoxicity effects towards RAW 264.7 cells with CC_50_ values > 500 μg/mL (Table 3). Therefore, at concentraetions ≤ 500 μg/mL, the extracts would not affect the host cells in plaque assays.

### 2.3. Anti-Viral Activity of HP, HWE, and HEE

We tested the dose-dependence of HP, HWE, and HEE on the reduction of viral titers (Table 4). The anti-noroviral activities of HP, HWE, and HEE were assessed by plaque assay (plaque forming units (PFU)/mL). *H. cordata* water extract (HWE) was assayed for its anti-viral activity on mouse norovirus 1 (MNV-1) at high (8.09 log10 PFU/mL) and low (4.38 log10 PFU/mL) counts following incubation with different concentrations (100, 250, and 500 μg/mL) of HWE for 30 min. When MNV-1 was treated with 100 μg/mL HWE, the viral titer was reduced. The amount of MNV-1 titer decreased by 1.38 log10 PFU/mL at high counts and by 1.87 log10 PFU/mL at low counts when compared to the untreated controls. At HWE > 250 μg/mL, MNV-1 titers decreased to undetectable level for low counts (4.38 log10 PFU/mL). Mouse norovirus 1 (MNV-1) was less sensitive to HP than HWE. At low counts, we obtained a greater reduction in viral titers with HP than at high counts. When MNV-1 was treated with 500 μg/mL HP, the infectivity of MNV-1 was decreased to undetectable levels in low counts. Anti-viral activity was also observed with HEE. At 100, 250, and 500 µg/mL, HEE was effective in reducing MNV-1 titers in a dose-dependent manner when compared to the untreated controls. However, when HEE was used with high and low viral titers, we obtained less than 1 log10 PFU/mL reduction in MNV-1 titer. The antiviral activity of *H. cordata* extracts was also evaluated by measuring EC_50_ (Table 3). Mean EC_50_ values were 187 μg/mL for HP, 76 μg/mL for HWE, and 1095 μg/mL for HEE. Mouse norovirus 1 (MNV-1) was more sensitive to HP and HWE than to HEE.

### 2.4. Time-Dependence Experiment of HP, HWE, and HEE on Viral Count Reduction

We tested the time-dependence of HP, HWE, and HEE on MNV-1 titer reduction. We incubated 250 μg/mL HP, HWE, or HEE with high counts of MNV-1 for up to 60 min and measured the viral titers. As shown in Figure 4, MNV-1 titers in PBS (untreated control) did not appear to change during the 60-min exposure, while titers in 250 μg/mL HP or HWE decreased significantly. The antiviral action of HP or HWE was rapid against MNV-1. Considerable changes (1.66 or 2.40 log10 PFU/mL) in MNV-1 titers were obtained in the first 10 min following HP or HWE addition. The counts continued to drop by 1.74 log10 PFU/mL for HWE and 1.53 log10 PFU/mL for HP for the next 30 min and continued to drop for the following 30 min. We obtained a total reduction in MNV-1 titers of 4.54 log10 PFU/mL for HWE and 3.69 log10 PFU/mL for HP. A time-dependence of the anti-viral activity of HEE on high titer MNV-1 was also observed. The viral titers of MNV-1 treated with HEE dropped slower in the first 30-40 min than HP and HWE and a total reduction in MNV-1 titers of 0.90 log10 PFU/mL for HEE treatment were achieved. HWE and HP had stronger antiviral effects on MNV-1 than HEE.

### 2.5. Viral Inactivation Kinetics

Table 5 shows that there was a 4.34- and 18.34-fold decrease in *D*-values with 250 and 500 μg/mL HP, respectively, compared to 100 μg/mL HP. We obtained a 1.0-log reduction with 100 μg/mL HP-containing samples at 29.09 min and a 1.0-log reduction with 500 μg/mL HP-containing samples at 1.59 min.

### 2.6. Effect of HP on MNV-1 Viral Particles

To further investigate the antiviral action of HP, the morphology of MNV-1 in low titer was analyzed by transmission electron microscopic (TEM) before and after treatment with PBS (0 μg/mL) and HP (250 μg/mL). Before treatment, MNV-1 particles were spherical (Figure 5A) and the size ranged from 30 to 35 nm as described previously [20]. After treatment with HP (250 μg/mL), the size of the MNV-1 particles increased to 80–100 nm in diameter. Additionally, we observed enlarged, denatured particles and disrupted particles (Figure 5B). The log10 reductions of MNV-1 under these conditions were 2.75 with 250 μg/mL HP. These results suggest that HP denatures the viral capsid proteins, thereby preventing viral adhesion.

## 3. Discussion

Although most studies have focused on investigating the role of *H. cordata* extracts against microorganisms [21,22], few studies have evaluated the effectiveness and mechanism of action of *H. cordata* extracts against HuNoVs. In this study, *H. cordata* extracts prepared in water (HWE and HP) and ethanol (HEE) exhibited various degrees of antiviral activity in a dose- and time-dependent manner. The values of selectivity index (SI) were 1.95 for HEE, 5.74 for HP, and 16.14 for HWE. Samples with high SIs had maximum antiviral activity, minimal cell toxicity, and a wider range of applications [23]. Among the extracts, HWE, mainly composed of carbohydrates, exhibited the strongest antiviral activity with the highest SI value. According to previous studies, carbohydrates are the main active substances in *H. cordata* water extracts and exhibited potent anti-viral activities [1,2,6,12,13]. In this study, we analyzed the *H. cordata* polysaccharide structures and their anti-noroviral mechanisms.

A polysaccharide (HP), isolated and purified from HWE with a molecular weight < 50 kDa, exhibited stronger anti-noroviral activity with higher SI value than HEE. Structural analysis revealed that HP mainly consists of α-1,4-linked Gal*p*A, β-1,4-linked Gal*p*, β-1,4-linked Glc*p*, and β-1,4-linked Xyl*p* residues. The predominant number of Gal*p* and Gal*p*A residues suggests that HP might be a pectin-like acidic polysaccharide with a 1,4-linked Gal*p* core. In addition, 1, 6-linked Gal*p*, 1,4,6-linked Gal*p*, 1,2,4-linked Rha*p*, 1,5-linked Ara*f*, and terminal-linked Gal*p* and Glc*p* in the side chains were detected. It has been proposed that some carbohydrates consisting of 1,4-linked GalpA may prevent the adhesion of some enteric pathogens such as protozoa, bacteria, and virus [24,25]. The polysaccharide (HP) is mainly composed of carbohydrates consisting of 1,4-linked GalpA and 1,4-linked Galp. The pectin-like acidic polysaccharide HP with a 1,4-linked Galp core might be the active substance in HWE responsible for the antiviral activity against MNV-1.

The Weibull model was used to evaluate the antiviral behavior of HP during 60 min (Appendix A). Based on the survival curves of MNV-1, the Weibull model showed the best fit to HP treatment with the highest R^2^ (>0.90) and lowest RMSE (<0.05) values. The kinetic model for the inactivation of MNV-1 can be analyzed with the *D*-value, which represents the time required to reduce the population of pathogens by 90% and can be measured from a time versus log survivors’ curve [26,27]. In our study, *D*-values decreased with increasing HP concentration. The *D*-values for MNV-1 inactivation were different (*p* ≤ 0.05) between the low treatments (100 μg/mL) and high treatments (250 and 500 μg/mL).

To further understand the antiviral mechanism of HP, we investigated the effects of HP on high-titer MNV-1 (8.09 log10 PFU/mL) using dose- and time-dependence experiments. Additionally, we observed the morphology of MNV-1 in low titer (4.38 log10 PFU/mL) with TEM under conditions that reduced its infectivity by more than 3 log10 PFU/mL. It has been reported that some polysaccharides inhibit viral infections by blocking the adsorption, entry, and/or cell-to-cell transmission of viruses. In addition, polysaccharides bind to viral envelope glycoproteins and disrupt them with negatively charged carboxylate groups, therefore minimizing/preventing viruses from penetrating target cells [28,29]. In our study, the changes in size and morphology of MNV-1 particles following HP treatment demonstrated that the polysaccharide had anti-norovirus activity by denaturing and enlarging virus particles to inhibit virus penetration. The morphological changes in MNV-1 following HP treatment were similar to those obtained in bacteriophage T4 and rotavirus with flavonoid compound CJs (cranberry juices) [30] and in MNV-1 treated with polyphenolic compound RCS-F1 (raspberry seed extract fraction-1) [31]. These results suggest that these natural compounds inactivate viruses by a similar mechanism.

In this study, we identified the anti-norovirus potential of HWE, HP, and HEE using a plaque assay for MNV-1. The pectin-like acidic polysaccharide HP, with 1,4-linked Gal*p* core, might be the active substance in HWE responsible for the antiviral activity against MNV-1. Currently, the only method for preventing noroviral infections is hand washing. In this study, the antiviral effects of HP and HWE reduced the residual MNV-1 infectivity following 10 min of incubation; therefore, these extracts interact immediately with the virus. The use of these extracts entails no safety concerns because *H. cordata* is a permitted food additive in China [32]. Our findings support the use of HP and HWE as nontoxic agents. Zhu et al. [33] and Kumar et al. [34] have shown that *H. cordata* exerts strong effects against a large number of enveloped and non-enveloped viruses. Therefore, HP and HWE might be potential antiviral agents in the prevention of viral diseases. However, we used unpurified HWE in this study. It is possible that total phenolics, proteins, and flavonoids in the extract may participate in the inactivation of the MNV-1 in a synergistic manner. It has been reported that a new type of flavonoid (Houttuynoids A–E (1–5)), from the whole plant of *Houttuynia cordata*, exhibited potent anti-HSV (herpes simplex viruses) activity [35]. The determination of the phenolic content in HWE through Folin–Ciocalteu solely corresponds to an estimation of the presence of reducing compounds in this study. Thus, future studies should investigate the characterization of polyphenolic profiles in HWE, and their mechanism of antiviral action.

## 4. Materials and Methods

### 4.1. Preparation of HP, HWE, and HEE

*Houttuynia cordata* was obtained from Zhejiang Chinese Medical University. Stems and leaves were air-dried at room temperature and ground (particle size ≤ 80 mesh) using a Thomas–Willey milling machine (Thomas Willey Mills, Swedesboro, NJ, USA). To prepare HEE and HWE, we followed the method reported by Sekita [36], with some modifications. To prepare HEE, we heated *H. cordata* samples (50 g) in an electromagnetic cooker at 450 W for 10 min, wrapped them in aluminum foil, and mixed them with 50 mL ethanol for 15 min. Following centrifugation at 2000× *g* for 15 min, the supernatant was recovered, lyophilized, and stored at 4 °C. To prepare HWE, *H. cordata* (50 g) was kept overnight at 16 °C, immersed in 200 mL distilled water, and boiled under traditional reflux for 90 min. After centrifugation at 2000× *g* for 15 min, the supernatant was recovered and concentrated under reduced pressure. Approximately 50 mL of concentrated extract was filtered, lyophilized, and stored at 4 °C. *H. cordata* polysaccharide (HP) was purified from HWE as previously reported [37]. Briefly, four volumes of 96% alcohol were added to HWE and stirred at room temperature. The precipitate was centrifuged at 8000 rpm for 15 min, washed several times with ethanol and acetone, and dried at room temperature. The purified polysaccharide obtained by DEAE chromatography (2.5 × 30 cm, Bio-Rad, Richmond, CA, USA) was filtered through Sephacryl S-300 (1.6 × 100 cm). The final fraction, labeled HP, was desalted and lyophilized. The homogeneity and molecular weight of HP were determined by HPLC (Waters, Alliance 2695 pump, Milford, MA, USA) coupled with a differential refractometer.

### 4.2. Chemical Analysis

The total carbohydrate content in *H. cordata* extracts was measured by the phenol–sulfuric acid method [37]. Phenolic compounds were expressed as mg of gallic acid/g (dry wt) of extract and tested by the Folin–Ciocalteu method. Protein was analyzed by the Lowry method. Uronic acid was measured by the *m*-hydroxydiphenyl method using galacturonic acid as the standard [38]. Total flavonoids were determined by the aluminum chloride method and expressed as mg of rutin equivalents (RE) per gram of extract [39]. In addition, monosaccharide composition was measured by 1-phenyl-3-methyl-5-pyrazolone (PMP)-HPLC [40]. Infrared spectra of HP were determined using KBr disk method and recorded at 400–4000 cm^−1^.

Methylation analysis was performed as previously reported [37]. Hydroxyl groups were methylated using lithium dimethylsulfonyl as anion and confirmed by FTIR spectroscopy. Methyl esters of uronic acids were reduced by lithium triethylborodeuteride (Superdeuteride^®^, Aldrich, Milwaukee, WI, USA) [38]. Methylated polysaccharides were subsequently hydrolyzed with 2 M trifluoroacetic acid for 2 h at 120 °C. Prior to analysis, the derivatives were reduced by NaBD4 and acetylated with acetic anhydride (Ac2O) and 1-methylimidazole (1-MeIm). Gas chromatography-mass spectrometry (GC/MS) was performed using an HP-5890 system coupled to an OV1701 column (0.25 mm × 30 m) and a temperature program ranging from 140 °C to 280 °C at 3 °C/min. The quantification of alditol acetate was carried out by the response factor and peak area of FID in GC. For nuclear magnetic resonance (NMR) analysis, the purified HP (60 mg/mL) was deuterium-exchanged by freeze-drying three times and then dissolved in 0.5 mL of 99% D_2_O. The spectra were recorded by a 500-MHz Bruker Avance 500 spectrometer for ^1^H and 125 MHz for ^13^C [41]. Signals at δ H 2.22 and δ C 31.1 for acetone were used as external standards.

### 4.3. Propagation of Viruses

Murine norovirus-1 and RAW264.7 cells were obtained from American Type Culture Collection (ATCC). Murine norovirus-1 propagation was performed by inoculation of confluent monolayers of RAW 267.4 cells. The RAW 264.7 cells were grown at 37 °C in a 5% CO_2_ atmosphere in Dulbecco’s modified Eagle medium (DMEM) supplemented with 10% heat inactivated 1× Anti-Anti (Gibco, Grand Island, NY, USA) and fetal bovine serum (FBS). Stocks of MNV-1 were prepared and stored at −70 °C [42,43].

### 4.4. Antiviral Effects of HP, HWE, and HEE

Ten-fold serial dilution of filter-sterilized *H. cordata* extracts (HP, HWE, and HEE) were used to determine cytotoxicity. Cytopathic effects were assessed by visual inspection after 3 d of incubation [44]. The cytotoxicity and virus-induced cytopathic effects were measured by MTT (3-[4, 5-dimethylthiazol-2-yl]-2, 5 diphenyl tetrazolium bromide) assay [45]. The 50% cytotoxicity concentration (CC_50_) value was determined following a 24-h treatment of RAW 264.7 cell monolayers to different doses of *H. cordata* extracts [18,45]. *H. cordata* extracts (HP, HWE, and HEE) were dissolved in sterile PBS at concentrations of 3000 μg/mL, sterilized by filtration (0.2-µm pore-size membrane), and diluted aseptically to 200, 500, 1000, 1500, and 2000 μg/mL in sterile PBS. To evaluate dose dependence, an equal volume of HP, HWE or HEE was added to an equal volume of MNV-1 resulting in titers of 8.09 log10 PFU/mL or 4.38 log10 PFU/mL and incubated for 30 min at room temperature. In this study, the 50% effective concentration (EC_50_) value was considered to be the concentration of *H. cordata* extracts required to reach only 50% of cytopathogenic effects caused by the MNV-1. Selectivity index was determined by the effectiveness at inhibiting MNV-1-induced cell death (CC_50_/EC_50_) [45]. For an analysis of time dependence, a set concentration (250 μg/mL) of HP, HWE or HEE was added to an equal volume of MNV-1 and incubated for 0, 10, 20, 30, 40, 50 or 60 min at room temperature. 2-thiouridine was used as a positive control. For the untreated control, we used individual viruses mixed with PBS. Following incubation, the reaction in each mixture was terminated with the addition of PBS.

### 4.5. Plaque Assay

Infectivity of MNV-1 was determined by the standardized plaque assay [10]. The RAW 264.7 cells were plated in 6-well plates at 2 × 10^6^ cells per well and grown until reaching 85% to 90% confluency. Serial ten-fold diluted viral strains were prepared in DMEM supplemented with 10% FBS. The viral dilution (0.5 mL) was inoculated into each well after aspiration of the media and incubated at 37 °C for 2.5 h. Following viral adsorption, the supernatant was removed and replaced with fresh DMEM (2 mL) containing 1% penicillin–streptomycin, 0.75% agarose, and 10% FBS. After 72 h of incubation, plates were overlaid with media containing 0.02% neutral red. Finally, plaques were calculated following incubation for 5 h at 37 °C. Titer reductions were measured by subtracting the titer of the samples from the titer of the PBS control.

### 4.6. Transmission Electron Microscopy (TEM)

Three-microliter aliquots of MNV-1 suspensions with *H. cordata* polysaccharide HP were diluted 10-fold in water and placed on a carbon-coated electron microscopy (EM) grid. The viral samples were stained with 3% aqueous uranyl acetate (3 μL) for approximately 1 min, air-dried, and observed under a JEOL-1400 electron microscope (Jeol, Tokyo, Japan) at 80 kV. Images were captured with a Gatan UltraScan 1000 XP camera (Gatan, Pleasanton, CA, USA) at a magnification of × 40,000 [46].

### 4.7. Viral Inactivation Kinetics

Viral inactivation kinetics were determined as previously described [26], with a slight modification. Briefly, HP solutions were prepared in PBS at 0, 100, 250, and 500 μg/mL HP. The polysaccharide (100 µL) was added to an equal volume of MNV-1. Following incubation for 0, 5, 10, 20, 30, and 60 min at room temperature, viral counts were measured by calculating the PFU titers. All experiments were carried out in triplicate. The decimal reduction time (*D*-value) was measured as previously reported [27] with slight modifications. The non-linear model (Weibull) can be expressed as Equations (1) and (2) [27].(1)$$\log_{10}\left(\frac{N}{N_{0}}\right)=-Kt^{\alpha }$$
(2)$$D={\left(\frac{1}{K}\right)}^{\frac{1}{\alpha }}$$
where *α* is the shape parameter, *K* is the characteristic time (min), and *D* is the time required to reduce the population of pathogens by 90%. The model-fitting ability was assessed by measuring the root mean squared error (RMSE) and the coefficient of determination (R^2^). To estimate the viral inactivation kinetics, the model was fitted using non-linear regression by Microsoft Excel 2013.

### 4.8. Statistical Analyses

Data were analyzed by Student’s *t*-test using the SPSS program. All experiments were carried out three times. Data were expressed as mean ± SE. Statistical significance was set at *p* ≤ 0.05.

## Figures and Tables

**Figure 1 molecules-24-01835-f001:**
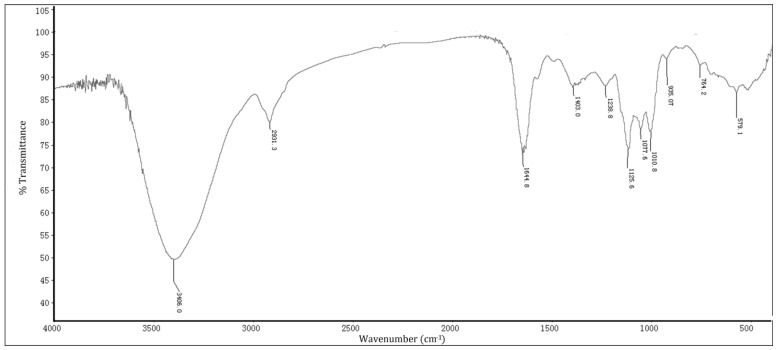
FTIR spectra of HP in the frequency range 4000–500 cm^−1^.

**Figure 2 molecules-24-01835-f002:**
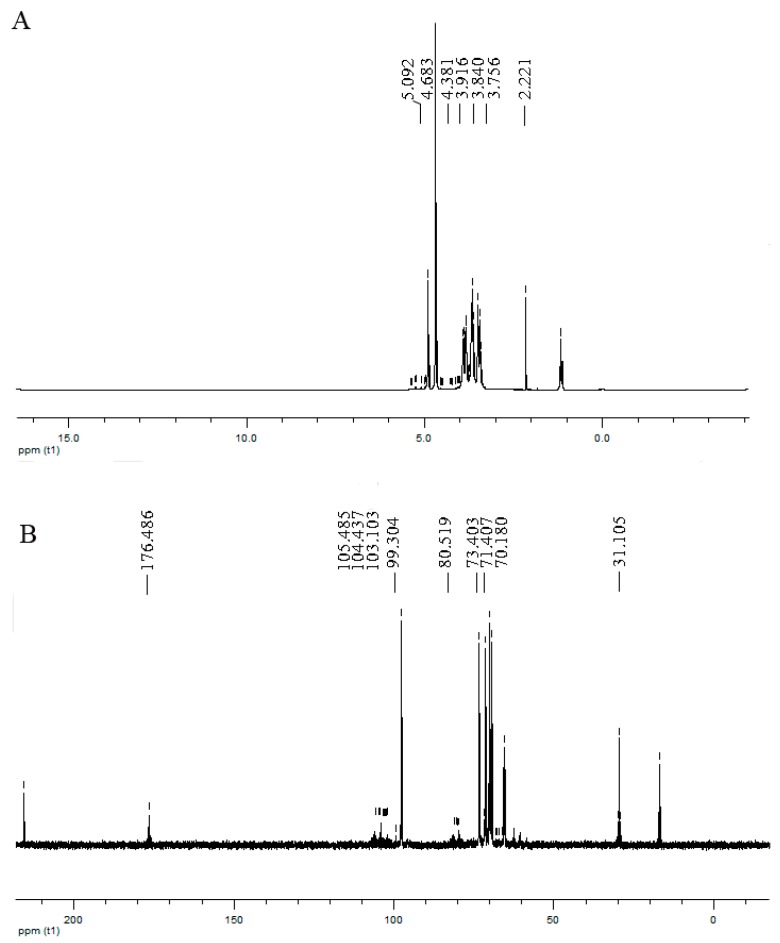
^1^H (**A**) and ^13^C (**B**) nuclear magnetic resonance (NMR) spectra of HP.

**Figure 3 molecules-24-01835-f003:**
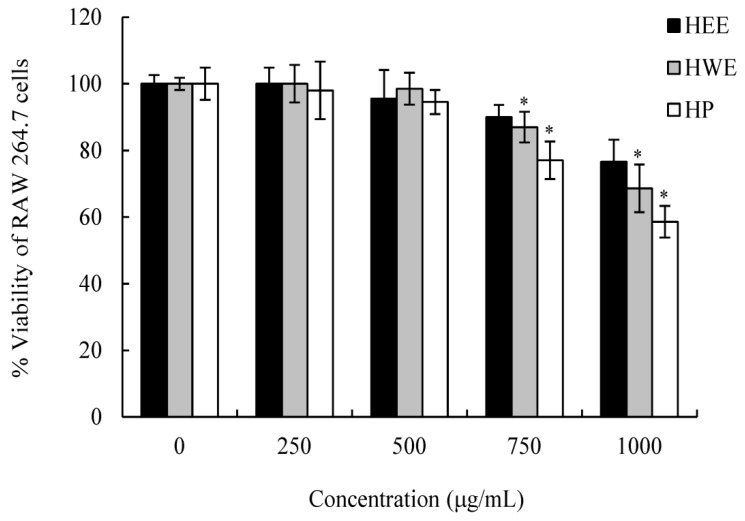
The effect of *Houttuynia cordata* extracts (HP, HCWE, and HCEE) 250, 500, 750, and 1000 μg/mL and phosphate-buffered saline (PBS) control (0 μg/mL) on the viability of the RAW 264.7 cell lines. Numbers are expressed as the % viability of bacteria cells and the murine macrophage cells (RAW 264.7) remaining after 60 min incubation with *Houttuynia cordata* extracts. A higher number of surviving reflects the concentrations of *Houttuynia cordata* extracts would not affect the ability to use these host cells to determine anti-viral effects using plaque assays. Each experimental condition was analyzed in triplicate. All data were expressed as mean ± standard deviation. Significant differences with HEE group are designated as * *p* ≤ 0.05.

**Figure 4 molecules-24-01835-f004:**
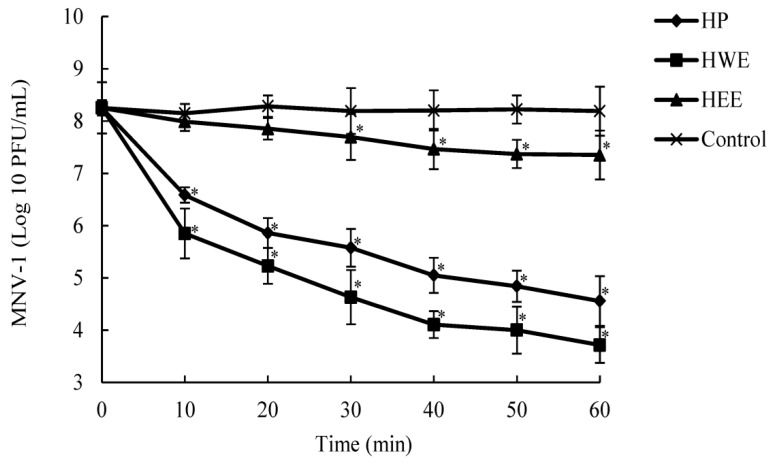
Change of MNV-1 titers versus incubation time after exposure to HP, HWE or HEE. Two-hundred-and-fifty (μg/mL) of HP, HWE or HEE solutions were added to an equal volume of MNV-1 at titer of ~8 log10 PFU/mL and incubated for up to 60 min at room temperature. The MNV-1 was recovered at 0, 10, 20, 30, 40, 50, and 60 min and assayed for its infectivity using standardized plaque assay. Each experimental condition was analyzed in triplicate. All data were expressed as mean ± standard deviation. Significant differences with control group are designated as * *p* ≤ 0.05.

**Figure 5 molecules-24-01835-f005:**
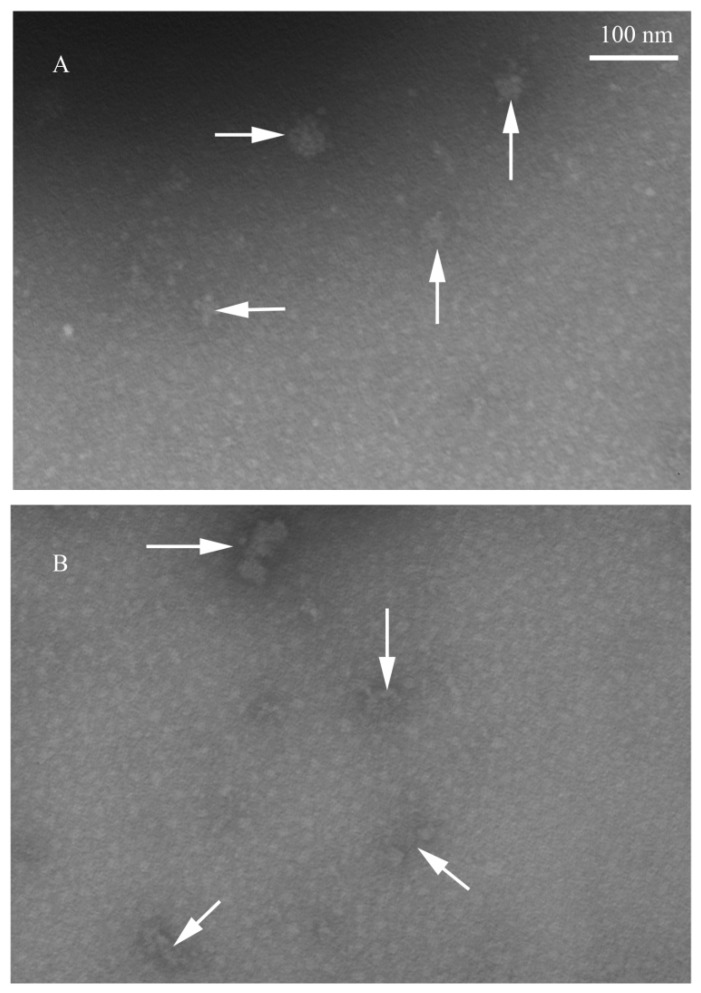
Transmission electron microscopic (TEM) images of MNV-1 in the absence or presence of HP. MNV-1 control (**A**) and MNV-1 treated with 250 μg/mL HP (**B**). Scale bars, 100 nm.

**Table 1 molecules-24-01835-t001:** Yields and chemical composition of *Houttuynia cordata* extracts HP, HWE, and HEE ^a^.

Composition ^b^	Extract/Fraction		
	HP	HWE	HEE
Extraction yield (%)	5.73 ± 2.16	16.31 ± 1.64	9.63 ± 2.31
Total phenolic (μg GAE/mg)	ND ^c^	14.59 ± 2.41	25.42 ± 3.17
Total flavonoid (μg RE/mg)	ND	10.62 ± 1.43	19.76 ± 3.73
Protein (%)	0.89 ± 0.12	2.54 ± 0.12	5.68 ± 1.76
Total carbohydrate (%)	81.12 ± 5.98	38.01 ± 3.98	2.37 ± 1.45
Neutral sugars (%)	56.33 ± 4.37	26.58 ± 2.54	2.37 ± 1.45
Uronic acid (%)	23.77 ± 3.42	12.34 ± 2.13	ND
Molar ratio of monosaccharides ^d^			
GlaUA	1.56	1.37	ND
Gal	1.49	0.94	ND
Rha	0.83	ND	ND
Ara	0.68	ND	ND
GluA	0.31	ND	ND
Glc	1.26	0.31	ND
Man	0.14	ND	ND
Xyl	1.11	ND	ND

^a^ HP, *Houttuynia cordata* polysaccharide; HWE, *Houttuynia cordata* water extract; HEE, *Houttuynia cordata* ethanol extract. ^b^ GAE, gallic acid equivalents; RE, retinol equivalent. ^c^ Not detected. ^d^ GlaUA, galacturonic acid; Gal, galactose; Rha, rhamnose; Ara, arabinose; GluA, glucuronic acid; Glc, glucose; Man, mannose; Xyl, xylose.

**Table 2 molecules-24-01835-t002:** Methylation analysis of *Houttuynia cordata* polysaccharide HP.

Sugar ^a^	Partially *O*-methylated Alditol Acetates	Molar Ratio	Linkage ^d^
Gal*p*A	2,3,6-Me_3_ Gal-6-d_2_ ^b^	1.74	→4)-Gal*p*A-(1→
Gal*p*	2,3,6-Me_3_ Gal	0.93	→4)-Gal*p*-(1→
	2,3,4-Me_3_ Gal	0.14	→6)-Gal*p*-(1→
	2,3-Me_2_ Gal	0.21	→4,6)-Gal*p*-(1→
	2,3,4,6-Me_4_ Gal	0.35	Gal*p*-(1→
Glc*p*	2,3,6-Me_3_ Glc	0.95	→4)-Glc*p*-(1→
	2,3,4,6-Me_4_ Glc ^c^	0.37	Glc*p*-(1→
Xyl*p*	2,3-Me_2_ Xyl	0.84	→4)-Xyl*p*-(1→
Rha*p*	3-Me Rha	0.22	→2,4)-Rha*p*-(1→
Ara*f*	2,3-Me_2_ Ara	0.17	→5)-Ara*f*-(1→

^a^ Gal*p*A, galactopyranosyluronic acid; Gal*p*, galactopyranose; Glc*p*, glucopyranose; Xyl*p*, xylopyranose; Rha*p*, rhamnopyranose; Ara*f*, arabinofuranose; ^b^ 2,3,6-Me_3_Gal-6-d_2_ = 1,4,5-tri-*O*-acetyl-6,6-dideutero-2,3,6-tri-*O*-methyl-galactitol. ^c^ 2,3,4,6-Me4-Glc = 1,5-di-O-acetyl-2,3,4,6-tetra-*O*-methyl-glucitol, etc. ^d^ Based on derived *O*-methylalditol acetates.

**Table 3 molecules-24-01835-t003:** Antiviral activities of *H. cordata* extracts against murine norovirus in RAW 264.7 cells using the mixed treatment assay.

*H. cordata* Extracts	CC_50_ (μg/mL) ^a^	EC_50_ (μg/mL) ^b^	SI ^c^
HEE	2132.43 ± 426.17	1095.53 ± 113.43	1.95 ± 0.84
HWE	1237.52 ± 367.65 *	76.75 ± 17.89 *	16.14 ± 3.81 *
HP	1074.76 ± 187.31 *	187.24 ± 77.82 *	5.74 ± 1.96 *

^a^ CC_50_: mean (50%) value of cytotoxic concentration. ^b^ EC_50_: mean (50%) value of effective concentration. ^c^ SI: selectivity index, CC_50_/EC_50._ Each experimental condition was analyzed in triplicate. Values are mean ± standard deviation. Significant differences with HEE group are designated as * *p* ≤ 0.05.

**Table 4 molecules-24-01835-t004:** Effects of HP, HWE, and HEE with low concentrations (4.38 log10 PFU/mL) and high concentrations (8.09 log10 PFU/mL) on mouse norovirus 1 (MNV-1) measured by plaque assay.

Extracts	Concentration	MNV-1 (log10 PFU/mL)			
		Low Count (4.38 log10)		High Count (8.09 log10)	
		Recovered Titer ^c^	Reduction ^e^	Recovered Titer	Reduction
PBS ^a^	0 μg/mL	4.38 ± 1.71	0	8.09 ± 1.95	0
2TU ^b^	50 μM	2.50 ± 0.83*	1.88	4.33 ± 2.14 *	3.76
HP	100 μg/mL	3.77 ± 1.04	0.61	7.49 ± 2.53	0.60
	250 μg/mL	2.62 ± 1.03 *	1.76	6.86 ± 1.81 *	1.23
	500 μg/mL	ND* ^d^	4.38	4.61 ± 1.72 *	3.48
HWE	100 μg/mL	2.51 ± 0.53 *	1.87	6.71 ± 2.12 *	1.38
	250 μg/mL	ND *	4.38	6.26 ± 1.32 *	1.83
	500 μg/mL	ND *	4.38	3.96 ± 0.74 *	4.12
HEE	100 μg/mL	4.21 ± 0.87	0.16	8.03 ± 2.89	0.06
	250 μg/mL	3.48 ± 1.33 *	0.90	7.52 ± 1.73	0.57
	500 μg/mL	3.24 ± 0.41 *	1.14	7.01 ± 2.07 *	0.98

^a,b^ Phosphate-buffered saline (PBS) was used as untreated control and 2-thiouridine (2TU) as a positive control. ^c^ Values are mean ± standard deviation. ^d^ Not detected. ^e^ Each titer was subtracted from the titer of the untreated sample (PBS). Each experimental condition was analyzed in triplicate. Significant differences with untreated control group are designated as * *p* ≤ 0.05.

**Table 5 molecules-24-01835-t005:** The Effect of HP with different concentrations on the *D*-values of MNV-1.

HP (μg/mL)	Weibull Model Parameters ^a^				
	*K* ± SD ^b^	α ± SD	*D*-Value (min) ± SD	RMSE	R^2^
100	0.144 ± 0.65	0.581 ± 0.04	28.098 ± 2.44	0.03	0.97
250	0.213 ± 0.01	0.817 ± 0.26	6.647 ± 0.93	0.02	0.98
500	0.866 ± 0.08	0.308 ± 0.04	1.597 ± 0.31	0.04	0.95

^a^*K* is the characteristic time (h); α = shape parameter; *D*-value = storage time (day) required to reduce MNV-1 or *Escherichia coli* by 90%; RMSE = correlation coefficient, a lower RMSE value indicates a better fit to the data; R^2^ = correlation coefficient, a higher R^2^ value indicate a better fit to the data. ^b^ Values are mean ± standard deviation. Each experimental condition was analyzed in triplicate.

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
