# Peer review of "Antiviral Effects of Houttuynia cordata Polysaccharide Extract on Murine Norovirus-1 (MNV-1)—A Human Norovirus Surrogate"

_molecules, 2019, doi:10.3390/molecules24091835_

Round 1

Reviewer 1 Report

This manuscript describes the antiviral effects of Houttuynia cordata polysaccharide against murine norovirus-1(MNV-1). This work is in follow up to previous paper (Carbohydrate Polymers, 2011, 83, 537-544) where they showed the antioxidant activities of an acidic polysaccharide and various solvent extracts from Houttuynia cordata. Although the experimental results support the author's conclusions, the data presented in the manuscript is not enough to conclude the claims. In addition, manuscript writing has lots of problem and mistake. Therefore, it seems insufficient to be published in this journal.

Comments:

1. Page 1, line 42: Thunb (italic) à Thunb. (normal)

2. In the sentence of line 79, page 2, please remove “novel”.  

Authors didn’t compare HP polysaccharide with previously isolated polysaccharides from Houttuynia cordata.

3. Please revise the sentence between line 114 and line 116, page 3.

4. Page 4, line145: 37 and CO2

5. This manuscript contained some poor English writing. Please check the manuscript for English grammar.

Ex) Lines 158 to 165 on the page 5, etc.

6. In the result, several contents didn’t match with figures or tables. Authors provided wrong figure and table numbers on many sites. (lines 188, 223, 228,230,235, 244, 252, 266, and 269)

7. Generally, IR data doesn’t suitable to propose the detailed structure such as galactose and glucose, etc. I suggest to remove or revise the content of lines 187~196 in the page 5.

8. In the Table 3, authors should describe for the term such as K, α, D-value, RMSE, and R2

9. In the Figure 4, TEM images didn’t give clear results. Please add pictures with high quality for accurate results.

10. Line 264 on page 8: were à was

11. For IR spectrum, transmittance is typically used for vertical axis.

Please use transmittance instead of absorbance in the Figure 1, and reverse upper and lower positions of the spectrum.

12. If authors are considered the polysaccharide as a novel one, authors need to provide all analysis data for the HP as a form of supporting information.

13. For antiviral activity against MNV-1, please add positive control data and EC50, CC50 and SI values for each sample.

Author Response

Response to the comments from reviewer 1

Comments:

1. Page 1, line 42: Thunb (italic) à Thunb. (normal)

Response: Thanks for reviewer’s comment. The change has been made according to the reviewer’s suggestion. (Line 45, Page 2)

2. In the sentence of line 79, page 2, please remove “novel”. Authors didn’t compare HP polysaccharide with previously isolated polysaccharides from Houttuynia cordata.

Response: Thanks for reviewer’s comments. In our study, the sugar composition of HP shows a strong similarity to the Houttuynia cordata polysaccharide fractions HCP and HCP-2 reported by Tian et al. (2011) [1] and Cheng et al. (2014) [2]. These comparing have been made according to the reviewer’s suggestion. (Line 93-95, Page 3)

3. Please revise the sentence between line 114 and line 116, page 3.

Response: Thanks for reviewer’s comment. The change has been made according to the reviewer’s suggestion. (Line 358-359, Page 12)

4. Page 4, line145: 37℃ and CO2

Response: Thanks for reviewer’s comment. The change has been made according to the reviewer’s suggestion. (Line 403, Page 13)

5. This manuscript contained some poor English writing. Please check the manuscript for English grammar.Ex) Lines 158 to 165 on the page 5, etc.

Response: Thanks for reviewer’s comment. We revised the whole manuscript with the help from a native speaker of English. We hope the reviewers can notice the improvement of the new version.

6. In the result, several contents didn’t match with figures or tables. Authors provided wrong figure and table numbers on many sites. (lines 188, 223, 228,230,235, 244, 252, 266, and 269)

Response: Thanks for reviewer’s comment. The wrong figure and table numbers have been rewritten according to the reviewer’s suggestion.

7. Generally, IR data doesn’t suitable to propose the detailed structure such as galactose and glucose, etc. I suggest to remove or revise the content of lines 187~196 in the page 5.

Response: Thanks for reviewer’s comment. The change has been made according to the reviewer’s suggestion. (Line 103-109, Page 3)

8. In the Table 3, authors should describe for the term such as K, α, D-value, RMSE, and R2

Response: Thanks for reviewer’s comment. The descriptions have been added according to the reviewer’s suggestion. (Table 5, Line 239-245, Page 9)

9. In the Figure 4, TEM images didn’t give clear results. Please add pictures with high quality for accurate results.

Response: Thanks for reviewer’s comment. The new TEM images have been added according to the reviewer’s suggestion. (Figure 5, Line 260-262, Page 10)

10. Line 264 on page 8: were à was

Response: Thanks for reviewer’s comment. The change has been made according to the reviewer’s suggestion. (Line 251, Page 9)

11. For IR spectrum, transmittance is typically used for vertical axis.Please use transmittance instead of absorbance in the Figure 1, and reverse upper and lower positions of the spectrum.

Response: Thanks for reviewer’s comment. The change has been made according to the reviewer’s suggestion. (Figure 1, Line 111-112, Page 3)

12. If authors are considered the polysaccharide as a novel one, authors need to provide all analysis data for the HP as a form of supporting information.

Response: Thanks for reviewer’s comment. The analysis data of GC-MS and (H,C) MNR have been added according to the reviewer’s suggestion. (Line 114-141, Page 4, 5)

13. For antiviral activity against MNV-1, please add positive control data and EC50, CC50 and SI values for each sample.

Response: Thanks for reviewer’s comment. The change has been made according to the reviewer’s suggestion. (Line 158-164, Page 6)

References

[1].  Tian, L.; Zhao, Y.; Guo, C.; Yang, X. A comparative study on the antioxidant activities of an acidic polysaccharide and various solvent extracts derived from herbal Houttuynia cordata. Carbohyd. Polym. 2011, 83, 537-544.

[2].  Cheng, B.H.; Chan, J.Y.W.; Chan, B.C.L.; Lin, H.Q.; Han, X.Q.; Zhou, X.; Wan, D.C.C.; Wang, Y.F.;Leung, P.C.; Fung, K.P.; Bik-San Lau, C. Structural characterization and immunomodulatory effect of a polysaccharide HCP-2 from Houttuynia cordata. Carbohyd. Polym. 2014, 103, 244-249.

Reviewer 2 Report

Introduction:

line 47: what is clearing heat?

line 51: delete "are"

line 79: revise the sentence. A full stop after H.Cordata interrupts the sentence which is meaningless.

Results:

lines 181-184: it is not necessary to report the molar ratio of monosaccharides, since they are reported in Table 1.

line 188: figure 1 instead of table 2.

line 244 and 252: probably it is Figure 3 instead of Figure 1 and Figure 2.

Lines 276-280: this short paragraph can be put before TEM, since the results concern the data reporetd in Table 3.

Can you provide some more details about the meaning of the parameter D, obtained from the fit using the Weibull model?

Author Response

Response to the comments from reviewer 2

1.       line 47: what is clearing heat?

Response: Thanks for reviewer’s comment. The change has been made according to the reviewer’s suggestion. (Line 49, Page 2)

2.       line 51: delete "are"

Response: Thanks for reviewer’s comment. The change has been made according to the reviewer’s suggestion. (Line 52-53, Page 2)

3.       line 79: revise the sentence. A full stop after H.Cordata interrupts the sentence which is meaningless.

Response: Thanks for reviewer’s comment. The change has been made according to the reviewer’s suggestion. (Line 77-79, Page 2)

4.       lines 181-184: it is not necessary to report the molar ratio of monosaccharides, since they are reported in Table 1.

Response: Thanks for reviewer’s comment. The change has been made according to the reviewer’s suggestion. (Line 91-94, Page 2, 3)

5.       line 188: figure 1 instead of table 2.

Response: Thanks for reviewer’s comment. The change has been made according to the reviewer’s suggestion. (Line 103, Page 3)

6.       line 244 and 252: probably it is Figure 3 instead of Figure 1 and Figure 2.

Response: Thanks for reviewer’s comment. The change has been made according to the reviewer’s suggestion. (Line 208, Page 8)

7.       Lines 276-280: this short paragraph can be put before TEM, since the results concern the data reporetd in Table 3.

Response: Thanks for reviewer’s comment. The change has been made according to the reviewer’s suggestion. (Table 5, Line 233-245, Page 9)

8.       Can you provide some more details about the meaning of the parameter D, obtained from the fit using the Weibull model?

Response: Thanks for reviewer’s comment. The details about the meaning of the parameters (such as D-value, K, RMSE and R2) have been added according to the reviewer’s suggestion. (Line 241-245, Page 9; Line 416-426, Page 14)

Reviewer 3 Report

With the current manuscript (molecules-467003), the authors report preliminary data on the antiviral effects of three uncharacterized extracts obtained from a mixture of stems and leaves of Houttuynia cordata. While the antiviral effects appear to be relevant, major issues, mainly concerning the chemical characterization of the extracts, preclude the acceptance of the current version of the manuscript. Below are my comments and suggestions.

Language and style editing are mandatory since the manuscript was written in an uncareful way, as in:

Lines 33-34:This suggests that the antiviral effects of the high carbohydrates component in H. cordata are probably related to its reduction of the inactivation of MNV-1.”. The sentence is unclear and scientifically incoherent.

Line 42: The taxonomic authority is (wrongly) italicized.

Lines 44-47:It was reported that H. cordata contains many constituents such as polysaccharides, fatty acids, polyphenols, flavonoids, sterols and other organic acids, and has various pharmacological activities including anti-viral, anti-cancer, eliminating toxins, clearing heat, anti-bacterial, and relieving stagnation [1].” This sentence is representative of the careless and sometimes scientifically inadequate writing that characterizes the whole manuscript.

Additional examples on the lack of organization are also in:

Line 188: “As shown in Table 2, the peak at…”. IR spectral data refers to Figure 1 and not Table 2.

Line 222: “…as shown in Table 3 Figure 3”.

Line 244: Figure 1 should be corrected to Figure 3.

Since the authors refer the isolation of a novel water-soluble polysaccharide (Line 79), further data allowing its structure elucidation should be provided. In fact, the chemical characterization of the polysaccharide is preliminary, authors solely providing data on the monosaccharide profiles, and a tentative elucidation based on IR data. Additional (and relevant) data on the branching points, glycosidic linkages and bond configuration can be easily obtained from methylation analysis, MS data or, preferably, from 2D NMR data, the authors being advised to consider it.

Furthermore, the authors solely provide an estimation on the phenolic and flavonoid content through more than limited approaches such as Folin-Ciocalteu. In fact, considering that the most pronounced effects towards MNV-1 virus were observed upon treatment with the aqueous extract, it is mandatory to provide further data on the possible compounds that may contribute to the antiviral effects, otherwise the current work does not present any scientific soundness. For example, it is worth mentioning that H. cordata is known as a source of potent anti-HSV-1 agents (doi: 10.1021/ol300017m), particularly flavonoids, that could possibly also contribute to the effects observed towards MNV-1.

While statistical analyses are described in Materials and Methods section (Line 166-171), the results on the cell viability (Figure 2) and virus count reduction (Figure 3), lack statistical data, not allowing to interpret the results correctly.

Authors are also advised to consider the Instructions for Authors from the current journal, since the manuscript does not follow the format indicated for research article sections (Results and Discussion should be included before Materials and Methods), references being also described in total disagreement with it.

It is my personal opinion that the results obtained with the current study are extremely preliminary, lacking scientific soundness, not being up to the journal standards.

Author Response

Response to the comments from reviewer 3

With the current manuscript (molecules-467003), the authors report preliminary data on the antiviral effects of three uncharacterized extracts obtained from a mixture of stems and leaves of Houttuynia cordata. While the antiviral effects appear to be relevant, major issues, mainly concerning the chemical characterization of the extracts, preclude the acceptance of the current version of the manuscript. Below are my comments and suggestions.

Language and style editing are mandatory since the manuscript was written in an uncareful way, as in:

      Response: Thanks for reviewer’s comment. We revised the whole manuscript with the help from        a native speaker of English. We hope the reviewers can notice the improvement of the new                version.

1.     Lines 33-34: “This suggests that the antiviral effects of the high carbohydrates component in H. cordata are probably related to its reduction of the inactivation of MNV-1.”. The sentence is unclear and scientifically incoherent.

Response: Thanks for reviewer’s comment. The sentence has been rewritten. (Page 1)

2.     Line 42: The taxonomic authority is (wrongly) italicized.

Response: Thanks for reviewer’s comment. The change has been made according to the reviewer’s suggestion. (Line 45, Page 2)

3.     Lines 44-47: “It was reported that H. cordata contains many constituents such as polysaccharides, fatty acids, polyphenols, flavonoids, sterols and other organic acids, and has various pharmacological activities including anti-viral, anti-cancer, eliminating toxins, clearing heat, anti-bacterial, and relieving stagnation [1].” This sentence is representative of the careless and sometimes scientifically inadequate writing that characterizes the whole manuscript.

 Response: Thanks for reviewer’s comment. The sentence has been rewritten. (Line 48-50, Page 2)

Additional examples on the lack of organization are also in:

4.     Line 188: “As shown in Table 2, the peak at…”. IR spectral data refers to Figure 1 and not Table 2.

Response: Thanks for reviewer’s comment. The change has been made according to the reviewer’s suggestion. (Figure 1, Line 103, Page 3)

5.     Line 222: “…as shown in Table 3 Figure 3”.

Response: Thanks for reviewer’s comment. The change has been made according to the reviewer’s suggestion. (Table 4, Line 179, Page 7)

6.     Line 244: Figure 1 should be corrected to Figure 3.

 Response: Thanks for reviewer’s comment. The change has been made according to the reviewer’s suggestion. (Figure 4, Line 208, Page 8)

Since the authors refer the isolation of a novel water-soluble polysaccharide (Line 79), further data allowing its structure elucidation should be provided. In fact, the chemical characterization of the polysaccharide is preliminary, authors solely providing data on the monosaccharide profiles, and a tentative elucidation based on IR data. Additional (and relevant) data on the branching points, glycosidic linkages and bond configuration can be easily obtained from methylation analysis, MS data or, preferably, from 2D NMR data, the authors being advised to consider it.

Response: Thanks for reviewer’s comment. The analysis data of methylation analysis, GC-MS and (H,C) MNR have been added according to the reviewer’s suggestion. (Table 2, Line 114-125, Page 4; Figure 2, Line 140-141, Page 5)

Furthermore, the authors solely provide an estimation on the phenolic and flavonoid content through more than limited approaches such as Folin-Ciocalteu. In fact, considering that the most pronounced effects towards MNV-1 virus were observed upon treatment with the aqueous extract, it is mandatory to provide further data on the possible compounds that may contribute to the antiviral effects, otherwise the current work does not present any scientific soundness. For example, it is worth mentioning that H. cordata is known as a source of potent anti-HSV-1 agents (doi: 10.1021/ol300017m), particularly flavonoids, that could possibly also contribute to the effects observed towards MNV-1.

 Response: Thanks for reviewer’s comment. H. cordata water extract (HWE) is mainly composed of carbohydrates in our study. Thus, the pectin-like acidic polysaccharide HP might be an active substance in HWE responsible for the antiviral activity against MNV-1. However, we used unpurified HWE in this study. It has been reported that   Houttuynoids A-E (1-5), a new type of flavonoid from the whole plant of Houttuynia cordata, exhibited potent anti-HSV (herpes simplex viruses) activity [1].Therefore, as mentioned above, it is possible that the amount of total flavonoids of H. cordata remaining in the extract also participate in the inactivation of the MNV-1 in a coordinated manner. To suggest the use of H. cordata extracts as natural remedies for foodborne viral illness prevention, further studies are required to identify the flavonoids of HWE and its mechanism of action against viral infectivity needs to be investigated. The discussion has been added according to the reviewer’s suggestion. (Line 324-328, Page 11)

While statistical analyses are described in Materials and Methods section (Line 166-171), the results on the cell viability (Figure 2) and virus count reduction (Figure 3), lack statistical data, not allowing to interpret the results correctly.

 Response: Thanks for reviewer’s comment. The statistical data have been added according to the reviewer’s suggestion. (Figure 3 and Figure 4)

Authors are also advised to consider the Instructions for Authors from the current journal, since the manuscript does not follow the format indicated for research article sections (Results and Discussion should be included before Materials and Methods), references being also described in total disagreement with it.

   Response: Thanks for reviewer’s comment. The change has been made according to the reviewer’s suggestion.

References

[1].  Chen, S.D.; Gao, H.; Zhu, Q.C.; Wang, Y.Q.; Li, T.; Mu, Z.Q.; Wu, H.L.; Peng, T.; Yao, X.S. Houttuynoids A–E, anti-herpes simplex virus active flavonoids with novel skeletons from Houttuynia cordata. Org. Lett. 2012, 14, 1772-1775.

Reviewer 4 Report

1- Line 70 reference needed to prove their oral administration safety. 

2- Line 81, It worth to remind the reader what are HP, HWE, HEE are refer to. 

3- Line 87-95 extraction methods used for HWE &HEE, please provide a reference and mention any optimisation.

4- table 3- give the reader more information in the table legend regarding the column title 

5- Figure 4A is not clear, it worth providing higher quality image to show the spherical shape that mentioned in the text.

6- More justification are needed in the discussion section.  

Author Response

Response to the comments from reviewer 4

1-       Line 70 reference needed to prove their oral administration safety. 

 Response: Thanks for reviewer’s comment. The references have been added according to the reviewer’s suggestion. (Line 70, Page 2)

2-       Line 81, It worth to remind the reader what are HP, HWE, HEE are refer to.

 Response: Thanks for reviewer’s comment. The change has been made according to the reviewer’s suggestion. (Line 77-78, Page 2)

3-       Line 87-95 extraction methods used for HWE &HEE, please provide a reference and mention any optimisation.

 Response: Thanks for reviewer’s comment. The references have been added according to the reviewer’s suggestion. (Line 335-343, Page 12)

4-       table 3- give the reader more information in the table legend regarding the column title 

Response: Thanks for reviewer’s comment. The information has been added according to the reviewer’s suggestion. (Table 5, Line 241-245, Page 9)

5-       Figure 4A is not clear, it worth providing higher quality image to show the spherical shape that mentioned in the text.

Response: Thanks for reviewer’s comment. The new figure has been added according to the reviewer’s suggestion. (Figure 5, Line 260-262, Page 10)

       6- More justification are needed in the discussion section.  

Response: Thanks for reviewer’s comment. The change has been made according to the reviewer’s suggestion. (Line 265-327, Page 10-11)

Round 2

Reviewer 1 Report

I read carefully this revised manuscript entitled “Antiviral effects of Houttuynia cordata polysaccharide extract on murine norovirus-1(MNV-1), a human norovirus surrogate”. Authors corrected well everything I had pointed out except for lines 104 – 110, page 3. Therefore, I think this work should be published in the Journal “molecules” after minor alterations as follows:

Authors provided several wrong IR data and contents.

1. Page 3, line 104: 3,406.0  à  3406.0

2. Page 3, line 105: 2,931.1 cm1 is attributed to C–H  à  2931.3 cm1 is attributed to sp3 C–H

3. Page 3, line 106: The peak at 1,644.8 cm1 might be due to C–O stretching vibration and/or N–H  à  The peak at 1644.8 cm1 might be due to amide C=O stretching vibration and/or C=C stretching vibration and/or N–H

4. Page 3, lines 107 and 108: The bands ranging from 1,600.6 cm1 to 1,420.0 cm1 reveal the presence of uronic acids, and the bands at 1,645, 1,514, and 1,403 cm1 are attributed to the presence of proteins  à  The bands at 1644.8, and 1403.0 cm1 are attributed to the presence of proteins

5. Page 3, lines 109 and 110: Moreover, the characteristic absorption at 1,077.6–899.7 cm1 is attributed to the presence of glycosidic linkages in HP [14].  à  Moreover, the characteristic absorption at 1200 – 1000 cm1 is attributed to the presence of glycosidic linkages C-O and/or C-N in HP [14].

6. Page 4, line 129: 1H spectrum  à  1H-NMR spectrum   

Author Response

I read carefully this revised manuscript entitled “Antiviral effects of Houttuynia cordata polysaccharide extract on murine norovirus-1(MNV-1), a human norovirus surrogate”. Authors corrected well everything I had pointed out except for lines 104 – 110, page 3. Therefore, I think this work should be published in the Journal “molecules” after minor alterations as follows:

Authors provided several wrong IR data and contents.

 Response: Thanks for reviewer’s constructive comments and suggestions for our manuscript.

1. Page 3, line 104: 3,406.0  à  3406.0

 Response: Thanks for reviewer’s comment. The change has been made according to the reviewer’s suggestion. (Line 103, Page 3)

2. Page 3, line 105: 2,931.1 cm−1 is attributed to C–H  à  2931.3 cm−1 is attributed to sp3 C–H

 Response: Thanks for reviewer’s comment. The change has been made according to the reviewer’s suggestion. (Line 104, Page 3)

3. Page 3, line 106: The peak at 1,644.8 cm−1 might be due to C–O stretching vibration and/or N–H  à  The peak at 1644.8 cm−1 might be due to amide C=O stretching vibration and/or C=C stretching vibration and/or N–H

 Response: Thanks for reviewer’s comment. The change has been made according to the reviewer’s suggestion. (Line 105, Page 3)

4. Page 3, lines 107 and 108: The bands ranging from 1,600.6 cm−1 to 1,420.0 cm−1 reveal the presence of uronic acids, and the bands at 1,645, 1,514, and 1,403 cm−1 are attributed to the presence of proteins  à  The bands at 1644.8, and 1403.0 cm−1 are attributed to the presence of proteins

 Response: Thanks for reviewer’s comment. The change has been made according to the reviewer’s suggestion. (Line 106, Page 3)

5. Page 3, lines 109 and 110: Moreover, the characteristic absorption at 1,077.6–899.7 cm−1 is attributed to the presence of glycosidic linkages in HP [14].  à  Moreover, the characteristic absorption at 1200 – 1000 cm−1 is attributed to the presence of glycosidic linkages C-O and/or C-N in HP [14].

 Response: Thanks for reviewer’s comment. The change has been made according to the reviewer’s suggestion. (Line 107, Page 3)

6. Page 4, line 129: 1H spectrum  à  1H-NMR spectrum   

   Response: Thanks for reviewer’s comment. The change has been made according to the    reviewer’s suggestion. (Line 131, Page 4)

Reviewer 3 Report

Despite considering previous suggestions and comments from the reviewers, the revised version of the manuscript molecules-467003 remains unsuitable for publication in the journal.

Firstly, extensive language and style editing is still required as evidenced in:

Lines 46/47: “…green leaves of H. cordata are popular vegetable products, and the leaves of H. cordata are being used in the preparation…”.

Lines 47-49: “… H. cordata contains a wide range of compounds including polysaccharides, fatty acids, polyphenols such as flavonoids, sterols, and other organic acids.” It is unclear which other “organic acids” are the authors referring to, since none of the previously mentioned classes of compounds correspond to organic acids. The sentence is scientifically incoherent.

Lines 64/65: “However, there are several disadvantages to each of these methods in food applications.” Consider “However, the use of such treatments is frequently inadequate in food industry”.

Lines 69:Pericarpium granati extract (PGE), and Pomegranate extract (PE),…”. Neither Pericarpium granati or Pomegranate should be italicized as they do not refer to any taxonomic designation.

Line 70: “These natural compounds are safe antiviral agents due to their health-conferring properties.”. Once again, the sentence is scientifically incoherent as “extracts” is not synonym of “compounds”. I am also unaware of any antiviral mechanisms based on “health-conferring properties”.

Numerous additional examples can be found throughout the whole manuscript.

Despite the additional spectroscopic data regarding the structural elucidation of the (potentially) new polysaccharide, the results obtained as well as the discussion remain insufficient. The discussion on the IR data is preliminary. For example, the peak at 1644.8 cm−1 might correspond both to amide C=O or C=C stretching vibration.

Concerning the NMR-based elucidation, the discussion is also extremely questionable as for example, the authors simply considered the resonances ranging from 4.8 to 5.24 ppm as indicative of α- and β- glycosidic bonds (which does not mean that much), not ascribing the values corresponding to the α- and β-anomeric protons. Authors are also requested to clarify what carbons are they referring to on lines 136-137, as no structure has been included on the manuscript. Detailed NMR data should also be included as supplementary materials since Figure 2 has a poor graphical quality, not allowing to interpret the data. Finally, it is extremely unlikely that 13C-NMR data has been obtained at 500 MHz (Lines 360-370).

On section 2.2., it is unclear what the EC50 values refer to. The results displayed on Figure 3 are also unclear, the authors being requested to clarify what bacterial cells? are they referring to.

As in my previous review report on the manuscript, I still fuel my opinion that the determination of the phenolic content through Folin-Ciocalteu solely corresponds to an estimation of the presence of reducing compounds (including for example carotenoids), and not exclusively phenolic compounds. As such, and fully agreeing with the authors (Lines 327-328), further studies on the characterization of the polyphenolic profile should be addressed, eventually providing further evidence on the bioactives responsible for the antiviral effects upon treatment with the aqueous extract (HWE)

Finally, as for the Introduction, the Discussion is messy, the first paragraph (lines 267-280) corresponding to a copy-paste version of the Introduction.

Considering the comments from the first review report as well as the ones abovementioned, it is my opinion that the manuscript should not be accepted for publication.

Author Response

Response to the comments from reviewer 3

Despite considering previous suggestions and comments from the reviewers, the revised version of the manuscript molecules-467003 remains unsuitable for publication in the journal.

Response: Thanks for reviewer’s constructive comments and suggestions for our manuscript. In this revised version, extensive changes have been made to address your concerns and to incorporate your suggestions. The changed sites were highlighted with red color. Please see the detailed response to reviewer’s comments. I hope that our manuscript has been improved and become acceptable for publication in the journal.

Firstly, extensive language and style editing is still required as evidenced in:

Lines 46/47: “…green leaves of H. cordata are popular vegetable products, and the leaves of H. cordata are being used in the preparation…”.

Response: Thanks for reviewer’s comment. The change has been made according to the reviewer’s suggestion. (Line 46, Page 2)

Lines 47-49: “… H. cordata contains a wide range of compounds including polysaccharides, fatty acids, polyphenols such as flavonoids, sterols, and other organic acids.” It is unclear which other “organic acids” are the authors referring to, since none of the previously mentioned classes of compounds correspond to organic acids. The sentence is scientifically incoherent.

Response: Thanks for reviewer’s comment. The change has been made according to the reviewer’s suggestion. (Line 47-48, Page 2)

Lines 64/65: “However, there are several disadvantages to each of these methods in food applications.” Consider “However, the use of such treatments is frequently inadequate in food industry”.

Response: Thanks for reviewer’s comment. The change has been made according to the reviewer’s suggestion. (Line 63, Page 2)

Lines 69: … Pericarpium granati extract (PGE), and Pomegranate extract (PE),…”. Neither Pericarpium granati or Pomegranate should be italicized as they do not refer to any taxonomic designation.

Response: Thanks for reviewer’s comment. The change has been made according to the reviewer’s suggestion. (Line 67-68, Page 2)

Line 70: “These natural compounds are safe antiviral agents due to their health-conferring properties.”. Once again, the sentence is scientifically incoherent as “extracts” is not synonym of “compounds”. I am also unaware of any antiviral mechanisms based on “health-conferring properties”.

Response: Thanks for reviewer’s comment. The change has been made according to the reviewer’s suggestion. (Line 69-70, Page 2)

Numerous additional examples can be found throughout the whole manuscript. Despite the additional spectroscopic data regarding the structural elucidation of the (potentially) new polysaccharide, the results obtained as well as the discussion remain insufficient. The discussion on the IR data is preliminary. For example, the peak at 1644.8 cm−1 might correspond both to amide C=O or C=C stretching vibration.

Response: Thanks for reviewer’s comment. The change has been made according to the reviewer’s suggestion. (Line 103-109, Page 3)

Concerning the NMR-based elucidation, the discussion is also extremely questionable as for example, the authors simply considered the resonances ranging from 4.8 to 5.24 ppm as indicative of α- and β- glycosidic bonds (which does not mean that much), not ascribing the values corresponding to the α- and β-anomeric protons.

Response: Thanks for reviewer’s comment. We apologized for the carelessness in the preparation of previous vision of ms. The NMR-based elucidation have been rewritten. (Line 129-145, Page 4)

Authors are also requested to clarify what carbons are they referring to on lines 136-137, as no structure has been included on the manuscript. Detailed NMR data should also be included as supplementary materials since Figure 2 has a poor graphical quality, not allowing to interpret the data.

Response: Thanks for reviewer’s comment. We apologized for the carelessness in the preparation of previous vision of ms. The detailed NMR data in Figure 2 and Table S1 have been added as supplementary materials in the revised version. (Line 129-145, Page 4)

Finally, it is extremely unlikely that 13C-NMR data has been obtained at 500 MHz (Lines 360-370).

Response: Thanks for reviewer’s comment. We apologized for the carelessness in the preparation of previous vision of ms. The NMR analysis in materials and methods section has been rewritten. (Line 355-359, Page 12)

On section 2.2., it is unclear what the EC50 values refer to. The results displayed on Figure 3 are also unclear, the authors being requested to clarify what bacterial cells? are they referring to.

Response: Thanks for reviewer’s comment. We apologized for the carelessness in the preparation of previous vision of ms. On section 2.2. The cytotoxicity assay was determined by calculating CC50 (mean (50%) value of cytotoxic concentration for RAW264.7 cells). On section 2.3. The antiviral activity of H. cordata extracts was also evaluated by measuring EC50 (mean (50%) value of effective concentration for MNV-1).The change has been made according to the reviewer’s suggestion. (Line 153, Page 5; Line 168-174, Page 6; Line 190-191, Page 7)

As in my previous review report on the manuscript, I still fuel my opinion that the determination of the phenolic content through Folin-Ciocalteu solely corresponds to an estimation of the presence of reducing compounds (including for example carotenoids), and not exclusively phenolic compounds. As such, and fully agreeing with the authors (Lines 327-328), further studies on the characterization of the polyphenolic profile should be addressed, eventually providing further evidence on the bioactives responsible for the antiviral effects upon treatment with the aqueous extract (HWE)

Response: Thanks for reviewer’s comment. The change has been made according to the reviewer’s suggestion. (Line 311-315, Page 11)

Finally, as for the Introduction, the Discussion is messy, the first paragraph (lines 267-280) corresponding to a copy-paste version of the Introduction.

Response: Thanks for reviewer’s comment. We apologized for the carelessness in the preparation of previous vision of ms. The first paragraph in discussion have been rewritten according to the reviewer’s suggestion. (Line 255-265, Page 10)
